# Sustainable Development and Crude Oil Revenue: A Case of Selected Crude Oil-Producing African Countries

**DOI:** 10.3390/ijerph17186799

**Published:** 2020-09-18

**Authors:** Ifeoluwa Adeola Ologunde, Forget Mingiri Kapingura, Kin Sibanda

**Affiliations:** 1Economics Sciences Department, Management and Commerce Faculty, Alice Campus, University of Fort Hare, Eastern Cape 5700, South Africa; 2Economics Sciences Department, Management and Commerce Faculty, East London Campus, University of Fort Hare, Eastern Cape 5700, South Africa; fkapingura@ufh.ac.za (F.M.K.); keith08.kin@gmail.com (K.S.)

**Keywords:** resource curse, paradox of plenty, unbalanced growth, pooled mean group, panel cointegration, autoregressive distributed lag

## Abstract

This study investigated the relationship between sustainable development and crude oil revenue (COR) in selected oil-producing African countries from 1992–2017 using the Pooled Mean Group (PMG) estimators on panel autoregressive distributed lag model (ARDL). Sustainable development was measured with the Human Development Index (HDI). This study was significant for Africa to break away from fiscal over-dependence on natural resource revenue, especially crude oil due to its high volatility and to correct porous institutional outlook. The a priori expectation is that crude oil revenue will tank so much that many countries will record negative positions and might not be to meet fiscal demands in the long run if the situation is protracted. Empirical results revealed that there was no long-term relationship between COR and sustainable development. In other words, the results suggest that any changes to COR have a potential negative effect on sustainable development in the selected countries. This implies over-reliance on COR will impact the economies negatively in the long run. This finding, therefore, requires an immediate fiscal intervention on spending on sustainable development drivers such as education, health, agriculture cum adoption diversification policy, and veritable supply-side policies that could avert the possibility of these negative effects and to correct traits of ineffective public institution. The absence of such policy interventions in these countries seems to be related to ineffective public institution and bad governance, culminating from poor, ineffective, and inefficient implementation.

## 1. Introduction

Today’s economies of developing nations seem to be challenged with more uncertainties than ever. Those requiring an immediate intervention are natural resource prices and the spate of ineffective policy formulation and implementation anchored on resource revenue, especially crude oil revenue (COR) within ineffective public institutions. They have been found to be unstable and unreliable when it comes to fiscal planning [1,2]. Surprisingly, many developing countries that are rich in crude oil build their economies around resource revenue anyway [3,4,5]. With the focus being crude oil revenue, which has been seen as the major natural resource revenue-driver across the world above other resources. Using current production capacity, top-ten crude oil-producing African countries were sampled, as in Table 1. It was observed that more than 70% of their GDP was crude oil-dependent [6]. This makes them susceptible to crude oil price volatility. This challenge, referred to as over-dependence on crude oil revenue, is further complicated by the basic truth that crude oil is non-renewable; depleting fast with population increase and production demands. So, the problem is hydra-headed and requires clear definition and status investigation to properly present strategic and feasible policy recommendations. Now, the objectives are very clear here. It has become more imperative for resource-rich developing nations of the world, especially those in Africa, to deal with these menaces of crude oil revenue (COR) overdependence and pursue a complete overhauling of the public institutions that will ensure attainment of sustainable development by driving indices of human development. Moreover, these ten nations’ productions expressed in 1000 barrels per day represented an average of 96.5% of all African crude oil production from 1992 to 2017 [7]. Additionally, the percentage contribution of the revenue generated from crude oil to gross domestic product (GDP) has not been less than 73% since the beginning of the 21st century at the latest [6,7]. Furthermore, each of these countries is very strategic to continental advancement. A good example is Nigeria. Apart from it being the most populated African country, it provides a huge market for fast-moving consumer goods, technologies, and agriculture. This is public knowledge. Other African countries being reviewed are scattered across the African continent in such a way that can easily promote multilateral trade relations. As is evident from Figure 1 below, which contains a map that pinpoints the distribution of crude oil deposits in the largest quantity and production in Africa, decisions on these countries with regards to sustainable development might impart Africa significantly. An understanding of how the revenues from this natural resource affect human development sustainably in terms of education, health, life expectancy, and gross national income per capita might be key to the much-needed African economic breakthrough.

In the past twenty-five years, the world has paid increasing attention to human sustainable development [8]. It will be inspiring to statistically investigate what has been happening in Africa in the light of crude oil revenue performance within the countries that garner it in abundance than the rest. It has been established that sustainable development cannot be achieved without sustained human development [8,9]. This is why the United Nations definition of sustainable development, “a process that meets present needs without compromising the ability of future generations to meet their own needs”, suggested that human development was and is still germane to achieving sustainable development [10,11]. This definition suggests that, in the face of limited resources, human well-being is in the heart of development and must be sustained from one generation to the other [11]. “It is bottom-up” [10]. A quick reference to “Agenda 21”, formed in 1992 at the Rio “Earth Summit”, to create action plans for sustainable development, is required to stress the need for measuring sustainability. This was why this study adopted the United Nations’ Human Development Index (HDI) as the measure of sustainable development.

The quantification tool of HDI has been used over and again, because it is a better metric for development than the likes of gross domestic product (GDP), adjusted net savings (ANS), green GDP, Wellbeing Index (WI), and many more [8,12,13]. Many of these measures of sustainable development do not capture the key elements that are contributory to human development as HDI does. HDI captures opportunities and needs to which humans always attach values. Such opportunities and needs such as education, health, and income are akin to sustainable development [11,14]. This measure is also better due to data limitation and over-simplification of the reality of other measures, comparably [12,15,16]. Sustainable human development index (SHDI) would have been a more suitable metric, but it only was published a few months ago, and it was applied to a single country with a gross lack of historical data for a panel study [6].

This study has become necessary, first, because of the macro-attention from which it has been empirically investigated across various researchers. More importantly, the statistics coming from Africa, especially the top ten crude oil-producing countries, have not been encouraging. In the world, over 1.2. billion are in extreme poverty, living below 1 US dollar per day. The majority of these people are in developing countries, with 6.5% in Eastern Asia, 24% in Sub-Saharan Africa, 44% in Southern Asia [17]. For example, in terms of economic performance, Nigeria, as an oil-producing country, has just recovered from a recession which was primarily triggered by the volatility of the world crude oil price, a major trait of resource curse [18]. The Democratic Republic of the Congo, Libya, Sudan, and Nigeria have all had to invest heavily in military equipment to curb civil wars and terrorism, which are other traits of resource curse [19,20,21]. Resource abundance seems to be a curse in developing countries, one can easily conclude. This has resulted in many oil-producing African countries being ranked in the lowest cadre of HDI [22]. Nigeria, Angola, Sudan, Congo, and Chad all rank among the lowest in the lower cadre of HDI ranking with 0.527, 0.516, 0.493, 0.418, and 0.396 respectively coming as 152nd, 150th, 165th, 176th, and 186th among 188 countries indexed and ranked by the UN in 2016. Corruption in terms of education fund embezzlement, contract diversion, political killings, porous and unreliable health system, and many more characterize many of these countries [23]. Inequality also stands at the highest among these countries with Congo having a Gini coefficient of 48.9 and Algeria ranking the lowest among them with 27.6, viewed relatively with their individual population.

The main objective of this study is to examine the contributions of crude oil revenue (COR) to sustainable human development in the selected crude oil-producing African countries. While doing this, the study sought to provide evidence that there are developmental and socio-economic roles crude oil revenue (COR) could play beyond contribution to the gross domestic product (GDP) of these countries. This is necessary to resolve the age-long paradox of plenty, popularly referred to as “resource curse” that has plagued these ten countries, considering that they contribute about 90% of African oil production [20,24,25]. This paradox has it that resource-rich African countries seem to have been more cursed than blessed with natural resources, especially crude oil [26,27]. Year-on-year, statistics have shown that the contributions of crude oil revenue (COR) to these countries’ GDP have never been less than 70% [20,24,28,29]. There has been much research and many publications carried out on how crude oil revenue (COR) imparts growth and development in these countries. These include [20,25,26,30,31,32]. Many of these publications are either more focused on its contributions to GDP and nominal expansion of the economies or on how it is compared in other world economies to determine comparative growth rate [33]. It is noteworthy that, careful investigation, no research has been conducted to understand how crude oil revenue (COR) in mono-economic African countries such as Nigeria, Angola, Gabon, and the selected rest, contribute to sustainable development using human development as a barometer within a panel study and compare it with other resource-rich African countries, thereby creating a workable template for other developing countries of the world. As earlier justified, HDI is particularly well suited to measure sustainable development regarding rent redistribution from resource sectors to non-resource sectors, which, if this process does not happen, gives birth to “resource curse”. This is expected to deepen the impacts of the resource rent across the economy when the redistribution translates into more employable and skilled graduates, improved primary and secondary health schemes, price stability, and stable and reliable institutions [30].

### 1.1. General Overview: Crude Oil Revenue and Sustainable Development

Oil revenue has been pivotal to economic development in many of the oil-rich African countries over the past five decades [34,35]. Using oil rent as a proxy for oil revenue, which is defined by Ravallion [29] as the difference between the value of produced crude oil and the total cost of its availability in the market, at world prices. It is expressed as a percentage of GDP because of its contributions. In terms of production, these selected countries ranked the highest among other oil-producing countries in Africa. They are the first ten according to the 2017 BP statistical review of world energy data. Figure 2 below shows how these countries compare with other oil-producing counterparts in Africa as of 2016.

At the world level, the World Bank has indicated that the oil revenue is not as reliable as it used to be for fiscal planning [23,36,37,38]. This is because of the fluctuations in the world oil price that continually cause unexpected shocks. To drill down to the selected countries, Figure 1 above shows a downward-sloping trend, indicated by a negative slope. The degree of responsiveness of change in oil revenue, in time, is clearly unreliable for fiscal planning in support of the World Bank’s position, even for the selected countries in Africa. This also supports the fact that it is not every time that the Dutch Disease rules correctly; it is its shocks, as pointed out by [38,39].

Figure 3 below presents a simple plot of oil rent-time trend coordinates across selected countries, being the top ten crude oil-producing African countries. Apparently, as opined by many apostles of resource curse, as a major setback to achieving sustainable development in resource-rich developing nations, oil revenue has been quite unstable for years in each of the countries under review. Looking closely at Figure 3, between 1992 and 2017, there were instances of unstable regional crude oil prices that plunged many of the crude oil revenue-dependent national budgets into deficit. Sometimes, unanticipated windfall gains, which were clearly transitory in nature, fulfilled some of the budgetary expectations. More clearly, Figure 4 below is an improvement on Figure 3 with the inclusion of the trend lines for each of the countries being reviewed.

Figure 4 indicated that oil rent exhibited positive trends in only Algeria, Chad, and Sudan between 1992 and 2017. In fact, a closer observation of the trend lines for these three countries showed that they were all elastic in behaviour; almost near perfect elasticity. This indicated that a proportional change in crude oil regional revenue or price was less responsive to a proportional change in time, ceteris paribus. This further confirmed the price elasticity of demand in Hotelling [37,38] alluded to by Gaitan et al. [39]. It plays out when used to measure the degree of revenue responsiveness to changes in price of crude oil. Demand is said to be inelastic if an increase in price will cause aggregate revenue to increase and vice versa [4,39]. Noteworthy is the fact that Algeria has the highest HDI ranks of all the ten countries being studied and the 83rd in the world among 188 countries as of 2015 World Bank’s rating and with similar crude oil production size of 1,348,361 million barrels per day as Angola and Nigeria, in the same year. The other two countries with positive oil rent trend ranked among the low HDI cadre in the world despite their massive daily crude oil production and comparably lower population as captured in Table 1 below:

Meanwhile, the remaining seven countries exhibited negative trends over the period of 1992 to 2017 as summarily captured by Figure 1 and Figure 2. Among all these countries, only Libya ranked better within the high HDI cadre as rated by the World Bank. Alarmingly, while Egypt, Gabon, and Equatorial Guinea were on medium HDI cadre, Nigeria, Angola, and Congo Republic all ranked among the low HDI cadre, despite having the first two of them as the highest producers of crude oil in Africa per day and also ranked 13th and 14th among highest crude oil-producing countries of the world respectively in 2015 according to OPEC ranking [22]. The questions remain, “why and what will this macroeconomic behaviour imply in the long run for sustainable development?” This is a significant reason for this study.

### 1.2. Literature Review

This study has its significance ingrained to the United Nations’ definition of sustainable development, which has its root in the Brundtland World Commission on Environment and Development (WCED) of 1983 [40]. Any description of sustainable development that does not lend credence to social, economic, and environmental pillars, might not be given the recognition expected. It, summarily, defines sustainable development as, “the principle for meeting human development goals while at the same time sustaining the ability of natural systems to provide the natural resources and ecosystem services, upon which the economy and society depend” [41]. Therefore, the bid to understand the contributory impact of oil revenue to development must revolve around variables that intrinsically define all the three pillars highlighted in the definition of sustainable development by the UN [42]. These are captured in Figure 5 below:

The United Nations (UN), as far back as 1990, was able to distinguish between an “end” and a “means” to an end [43]. Resources, natural or produced and exhaustible, as the case may be, cum the revenue they generate, never really sufficed in defining sustainable development [29]. The 1990 Human Development Report (HDR) specifically mentioned that, “Human beings are the real end of all activities, and development must be centred on enhancing the achievements, freedom, and capabilities”. “It is the lives they lead that is of intrinsic importance, not the commodity or income they happen to possess” [42]. Therefore, human development index captures the concepts of social, environmental and economic levels of development [9]. It is a composite index that captures the following [11]:➢average years of schooling➢years of schooling expected➢birth life expectancy➢the gross national income per capita

Clearly, this model was designed to measure how human existence is secured and sustained from birth through life. Three variables stood out and therefore congealed into the following Table 2:

Cockx and Francken argued in favour of HDI [44]. They believed one of the blind spots left unconsidered in understanding and resolving “resource curse” is the impact a natural resource revenue has on government spending. They opined that innumerable government officials misappropriate resource revenue for personal gains rather than drive growth of HDI for sustainable development [25]. Meanwhile, many authors have suggested alternative metrics to measuring sustainable development [12,13,23,45]. Such metrics include adjusted net savings (ANS), defined as the gross national savings net of depreciation of produced capital, plus education expenditure, minus natural resources rent and carbon dioxide emissions [13]. Others are sustainable budget index (SBI), green GDP, Openness Index (OI), system of environmental-economic accounting (SEEA), national accounting matrix including environmental accounts (NAMEA), etc. Many of them do not capture the social wellness of the population such as HDI, others are restricted to time and location, generally intended to depart from using GDP to measure sustainable development without paying attention to human development and wellbeing [13,23].

Also, Cockx and Francken [44], using a panel dataset, mentioned that revenue from natural resources, especially crude oil, provided a valuable source for growth and development for resource-rich countries. However, they found that, in a panel data on 140 countries, such countries have experienced slower achievement of sustainable development, as there was an inverse relationship between resource-dependence and education spending [44]. They also found out that gains from trading natural resources are still less distributed in developing resource-rich nations of the world compared with their developed counterparts. As a result of this, investment in education, health, and other human development areas are crowded out by natural resource windfall [44]. Odunsi [45] supported Cockx and Francken’s [44] findings as seen in a more recent UN report on human development. He found that several oil-rich developed countries such as Norway, Germany, Canada, the Netherlands, etc. were reported to rank among the top ten nations with the highest HDI. The reason was their capabilities to efficiently redistribute oil revenue to other complementary sectors to promote education, health, technology, and other sustainable development drivers [13]. Countries such as Canada, Norway, and Germany have a remarkable focus on developing sustainable human capital by creating a funded bridge between the industries and schools, hospitals, empowerment agencies, etc. at the lowest level of human contact in the society. They created sustained agricultural production and technological progress through resource rent diversification.

This was supported by Neumayer [13] in a study, using an efficiency decomposition method to measure the transformation of the minimum possible resource capacity of a country into maximum possible and efficient levels of outcome in form of improved life expectancy, education, and per capita income. In addition, Hayashi et al. [46] argued in a study to measure rural-urban disparity that Japan, created one of the most egalitarian societies by economically fortifying the rural regions to have access to what is obtainable in the cities, though it was an oil-importing country. It was found out that many people would rather stay in the rural area and earn a lower income than migrate to the cities and experience human and vehicular congestion and a comparably higher cost of living. This was supported by Randall and Nakamura [47] report on the subject matter. Since the concentration was on the crude oil-producing African countries, Table 3 below shows how top crude oil-producing African countries fared along with a few others in 2018 with regards to net export of crude oil. Among the top fifteen countries in the world, there are the top three crude oil-producing African countries namely Nigeria, Angola, and Libya, ranking 8th, 10th, and 12th respectively. The statistics below present the surplus between the value of each country’s crude oil exports and its import purchases for that same commodity.

These statistics further support the argument that crude oil revenue is key to achieving sustainable development in developing countries of the world if properly used.

Li [21] also supports the opinion that resource revenue does not change poor economies into flourishing ones except it was effectively, efficiently, and inclusively redistributed. Thus, supporting economic diversification [48]. In addition, Shao and Yang [4], in their normative study using an economic operating conceptual mechanism and a mathematical endogenous growth model, found out that sufficient and well-developed human capital was required for sustainable development with well-distributed oil revenue as a more dependable catalyst in oil-rich developed nations.

Li [21] carried out a normative study on the crude oil between Libya and Botswana. Li found that the latter’s economy has fared better in the absence of resource revenue. Additionally, Mohammad and Riyazuddin [49] found out that Saudi Arabia attainment of “very high HDI” status was largely dependent on well-diversified use of crude oil revenue. Literally, a 1% increase in oil production increased HDI by 4% points and a 1% increase in government spending led to a 10% increase in HDI points [49]. The heavy contribution of oil revenue to total government revenue from export revenue was supported by Sultan and Haque [50], who put it at 90%. Sultan and Haque [50] went further to test their assumptions of the existence of a long-run relationship between oil revenue (exports) and drivers of sustainable development on which the government spent. The results emerged with the acceptance that oil revenue (exports) was positive and significant to sustainable development using the Johansen cointegration test [50]. One of the major drivers of sustainable development is income expressed in the index ‘GNI per capita’ [22,48,50,51]. Alkhateeb, Sultan, and Mahmood [50] studied the relationship between oil revenue and employment, in a bid to understand how the former contributed to the latter. They found out from their results that increase in oil revenue was contributory and significant to GNI per capita growth, because wages and employment also increased in the process. This result was supported and expanded by Lorusso, Pieroni, and Lorusso [48], who studied and checked the “causes and consequences of oil price shocks” in the United Kingdom (UK). Lorusso et al. [48] discovered that an increase in oil price expanded the capacity of the UK government to drive sustainable development. This was because increased oil price led to increased revenue, which in turn reduced government deficit. Therefore, these empirical works showed differing results across regions. This further pointed out the importance of this study among the sampled countries.

It is important to identify frameworks that underpin the argument of this study. Such frameworks must provide context that pokes holes in the current mono-economic system in play and make clear allusions to sustainable development in one way or the other. Such proponents include Professor John Martin Hartwick, Warner Max Corden, and J. Peter Neary, Harold Hotelling, and Prof. A.O. Hirschman. These five selected contributed to development economics, and by implications, to petroleum, energy, environmental, resource, and health economics. The Hartwick’s Rule by Professor John Martin Hartwick, in 1976 suggested that sustainability can be achieved by reinvesting rent from exhaustible capital or non-renewable environmental resources to produce artificial capital, thereby making net investment zero [31]. In other words, when the state invests revenue earned on exhaustible resources, at a point in time, it should be done to create produced capital, both tangible and intangible, to secure the future [52]. Hartwick advocated for sustainability by ensuring the irreplaceable capital exploited from the environment at the current time is efficiently used such that generations to come will continue to depend on its remaining deposit. It advocated for intertemporal use of natural resources. In addition, the Dutch Disease framework describes the massive dependence on gain or rent from a booming natural-resource-driven sector, the macroeconomic structural adjustments (such as wage increase, appreciation of local currency, price volatility, etc.) that occur is summarily called the Dutch Disease. Especially because of its negative impacts on the local industries’ export sales value [21]. It is suited for examining the contribution of oil revenue from the “booming sector” of a resource-based economy, on development because of the various socio-economic inferences and impacts that it eventually revealed. In reality, most oil-producing African countries have been seen to experience various negative swings from low HDI, corruption, political violence, price volatility, weak institutions, etc. as pointed out in the model [53]. The model also pointed out that economic diversification and inclusion in countries that have been experiencing “resource curse” will help them attain sustainable development [21]. The Harold Hotelling framework was to address decision-making on the correct price on exhaustible natural resources. It is relevant here because it imposes prudency on economic stakeholders. It does not encourage inefficient extraction of non-renewable natural resources. Finally, the theory of unbalanced growth (TUG) was developed to resolve the problem of over-reliance on natural resource revenue [34,54,55]. This framework is appropriate to this study because it seeks to diffuse the gains of the booming sector into investing in a sector that has economic forward and backward linkages to other sectors. Hirschman [51,56] implied in his theory that a developing country can only invest in any of the following categories of investments:Social Overhead Capital (SOC)—services undertaken only by public agencies;Direct Productive Activities (DPA)—investments undertaken by private corporations in areas that add to the flow of final goods.

TUG suggested that the state should only undertake investments in either of the categories above but must identify the sectors with the highest forward and backward linkages before setting out. The linkage theory in the model of unbalanced growth is very relevant to the study on the effect of oil revenue on sustainable development with special reference to fiscal planning. Many researchers such as [6,21,36,37,57,58], have proposed revenue diversification for developing countries over time. However, very few of these countries have gotten it right. This speaks directly to the efforts geared at achieving sustainable development, even in the face of unreliable oil revenue and its shocks. Therefore, these frameworks appeal directly to the challenges culminating from the relationship between crude oil revenue and sustainable development in these developing nations.

Now, with all the arguments presented to show the ills of overdependence on crude oil revenue (COR), does it mean it is a necessary evil that has to be endured by countries with commercial abundance? Obviously not. Empirical results have shown that oil is not necessarily evil, but it can become weaponized through inept and power-drunk leadership, institutional porosity, corruption among public and private economic administrators, lack of fiscal continuity due to political instability, in-terrorism, and many more [3,39,40,41,42,43,44]. All these attributes have been identified as some of the reasons the growth and development of developing nations in Africa, Asia, and the Middle East have been sluggish.

Lee, Chang, Arouri, and Lee [23,56] also supported the negative impact crude oil and other natural resources on development through trade. Most resource-rich countries, especially in Africa, have been distracted from solving the weak trade openness created by resource revenue in a one-sided economy [56]. Lee, Chang, Arouri, and Lee [23,56], by using an innovative dynamic panel threshold model, assessed the strength of institutional environments in developing nations. They found that development is impeded because of an unhealthy institutional environment there. These impediments to stable, reliable, and credible institutions have been significantly tied to crude oil discovery and its huge revenue outlay. Contract lobbying, public fund misappropriation, and many more ills, according to [3,56,59,60], have taken negative tolls on sustainable development because of uncontrolled crude oil windfall. Specific studies drew this conclusion using various methodologies such as OLS, panel dataset, Johansen co-integration model, Multi-Criteria Decision-making (MCDA), etc. [6,7,11,43,58,61,62,63]. However, while some of them were country-specific studies, others were reviews. None of these literatures or other existing ones studied the contribution of crude oil revenue to sustainable development across the first ten oil-producing developing African countries, as selected in this study.

The developed and technologically advanced countries have been able to attain sustained human development that attracts brains and investments from other nations of the world without necessarily using crude oil or any natural resource revenue, but on the back of effective government institutions and private sector control, thereby recording favourable corruption indices. In addition, the studies revealed a consensus on the need to increase health and education investment, create a system devoid of corruption, and diversify the economy. Findings also showed that investment in education and health in developed countries are better and have been existing for longer when compared to developing middle-income countries. However, oil-importing developed nations have fared better in the attainment of sustainable development goals than their oil-producing counterparts. Examples of such countries are China, United States of America, South Korea, Japan, etc. However, Canada, Germany, Norway, and other developed oil-producing countries have significantly reduced dependence on oil revenue and have also created sustainable development by developing human capabilities to secure the future, irrespective of the size of oil reserves or deposit available [44,57,64].

### 1.3. Sustainable Development and CPIA

Experts have identified more factors that affect the effectiveness of resource deployment in developing countries. Such factors are corruption, institutional weakness, and policy ineffectiveness, The World Bank developed an index called “Country Policy and Institutional Assessment” (CPIA). This index allows a country to be rated against a group of sixteen criteria classified into four, which are economic management, structural policies, social inclusion and equity policies, and management of public sector and institutions. This index allows the rated countries and stakeholders to understand the extent of policy implementation transparency, accountability, and degree of corruption within the public sector. The study looks at the trend of CPIA so as to determine the effects of corruption on policy implementation in economies majorly enriched from crude oil revenue (COR), and other institutional assessment criteria needed to ensure those countries attain sustainable development and within a specified time. This index is also used to determine and classify the degree of fragility of economies of the world, especially those of developing African countries with huge market sizes and deposit of natural resources.

## 2. Materials and Methods

This study employs yearly panel data covering the period 1992 to 2017. The data for human development index (HDI), oil revenue, life expectancy, total public health expenditure, income per capital, and gross domestic product (GDP) were obtained from the World Bank data group database. Other data used for descriptive and conceptual analyses were obtained from British Petroleum Statistical Review of World Energy, 2017, and the central banks of the African countries reviewed. The natural logarithms of all the series were taken to curb the effect of imbalanced values, number sizes, and outliers. Meanwhile, the study modified a model by Mohammad and Md Riyazuddin [49] using PMG estimators on the panel autoregressive distributed lags model. Additionally, stacked columns charts were used to explain the impact a composite variable CPIA has on the attempts made at attaining sustainable development with respect to crude oil revenue (COR) in the selected countries. The data used for CPIA is available between 2005 to 2017 among five of the countries studied. In the panel analysis, the PMG estimator model has been seen as a good alternative to estimators such as DOLS and FMOLS. This is because PMG will present important evaluations about the presence of long-run homogeneity in the relationship between sustainable development and oil revenue. Therefore, the PMG estimator model will be applied in order to find the long- and short-run relationship between sustainable development and oil revenue. Meanwhile, Mohammad and Md Riyazuddin [49] argued that HDI was largely affected by oil revenue and government spending, with the latter also predominantly dependent on the same oil receipt. Mohammad and Md Riyazuddin [49] further argued that sustainable development is a protracted or prolonged current and subsequent economic development, provided the positive trend is maintained or grown. The authors used the ordinary least squares (OLS) estimation technique to investigate the roles oil production and government spending played in the improvement of the Saudi Arabians’ HDI rating by the UNDP [12]. Below is the statement of the functional OLS models estimated model by Mohammad and Md Riyazuddin [49]:(i)
HDI_t_ = *β*_0_ + *β_*1*t_* + *Oil production* + µ
(ii)
HDI_t_ = *β*_0_ + *β_*1*t_* + *government spending* + µ

The ARDL (*p*, *q*, *q*, …, *q*) model is specified in a generalised and re-parameterised form as follows, respectively:(1)$$Y_{it}=\overset{p}{\underset{j=1}{\sum }}\delta_{ij}Y_{i,t-j}+\overset{q}{\underset{j=0}{\sum }}{\beta^{\prime }}_{ij}\;X_{i,t-j}+\varphi_{i}+\varepsilon_{it}\;$$
(2)$$\Delta Y_{it}=\theta_{i}\left[Y_{i,t-j}-{\lambda^{\prime }}_{i}X_{i,t}\right]+\overset{p-1}{\underset{j=1}{\sum }}\xi_{ij}\Delta Y_{i,t-j}+\overset{q-1}{\underset{j=0}{\sum }}{\beta^{\prime }}_{ij}\;\Delta X_{i,t-j}+\varphi_{i}+\varepsilon_{it}\;$$
$Y_{it}$ is the dependent variable, $X_{it}$ is the vector representations that are permitted to be integrated of order I(0) or I(1) or cointegrated; $\delta_{ij}$ is the coefficient of the lagged explanatory variables referred to as scalars; $\beta_{ij}$ are vectors with K×1 coefficient; $\varphi_{i}$ is the unit-specific fixed effects; i=1, …, N;t=1, 2, …, T;p, q are optimal lag orders; $\varepsilon_{it}$ is the error term. is the $-(1-\delta_{i}$), is the group-specific speed of adjustments coefficient (expected that $\theta_{i}<0$); ${\lambda^{\prime }}_{i}$ is the vector of long run relationships and $\left[Y_{i,t-j}-{\lambda^{\prime }}_{i}X_{i,t}\right]$ is the error correction term (ECT), and $\xi_{ij}$, ${\beta^{\prime }}_{ij}$ are the dynamic coefficients of the short-run. To properly present the model specification, see Equation (3):(3)$$\Delta HDI_{it}=\theta_{i}\left[HDI_{i,t-j}-{\lambda^{\prime }}_{i}X_{i,t}\right]+\overset{p-1}{\underset{j=1}{\sum }}\xi_{ij}\Delta HDI_{i,t-j}+\overset{q-1}{\underset{j=0}{\sum }}{\beta^{\prime }}_{ij}\;\Delta X_{i,t-j}+\varphi_{i}+\varepsilon_{it}\;$$
$\Delta HDI_{it}$ is the dependent variable with the difference operator; Equation (3) was used to estimate the short-run and long-run relationship between sustainable development (with HDI as the proxy) and crude oil revenue (with Oil_rent as the proxy) in conjunction with other adopted predictors in the model. There are two steps for estimating a long-run relationship involved in this technique. First, is the investigation if the existence of a long-run relationship among all variables, which is referred to as cointegration. Once co-integration is confirmed among the variables, the second step will be to estimate the long-run coefficients in accordance with the results of the ARDL model [65]. Panel ARDL was chosen for this study based on the fact that panel data analyses involve the use of both cross-section (N) and time series (T) observations for analysis and allow for inclusion of lagged dependent variable as a regressor [66,67]. Now, cross-equation restrictions within the long-run parameters have to be implemented using maximum-likelihood estimation because the approach is being used in panel data. To provide the validity of the restrictions, the Hausman (1978) test [68] is used. Then, estimations are provided by the Pooled Mean Group (PMG) estimators.

Second-generation unit root tests were utilized to ascertain data stationarity. These are the Maddala and Wu tests. Maddala and Wu [43] proposed to solve the problem of panel unit root testing using p value combination tests. A non-parametric and exact test was proposed based on Fisher’s [69] test and combining the probability values of occurrence from every panel member’s unit root tests. In addition, the ARDL techniques, which allow the lags of the samples within it to be distributed fit these properties of an unbalanced panel. It is a model that determines the value of the dependent variable using the current coefficient of an explanatory variable and the lead value of the same variable [70,71]. The panel ARDL model allows for the variables to be either integrated of order I (0), I(1), or both [71,72,73]. Meanwhile, in the PMG ARDL analysis, the main interest is group analytics and not just the individual units in the group. Therefore, little or no information is lost by considering the panel perspectives. The use of panel data reduces the possibility of problems or noises such as heteroscedasticity. It also increases the number of observations and their variation. Panel ARDL is best suited for countries where data administration and warehousing pose as issues. It works well with an unbalanced panel. Important among its attributes is the heterogeneity of the cross-sectional units. Hence, it allows for subject-specific variables. Panel ARDL is also well suited for investigating dynamic variations culminating from repeated cross-sectional observations.

The decision rule is that:ect. ≥ −1 and ect. ≤ −2 (can be more than −1, but not lower than −2);No cointegration to long-run equilibrium if ect. > 0; the model is therefore explosive for such sample;*p*-value ≤ 0.05 and ect. coefficient is the adjustment speed to deviation from long-run equilibrium.

The next step was to determine more suitable estimators between the PMG and the MG using the Hausman (1978) test, as earlier mentioned. This is also called the Durbin–Wu–Hausman (DWH) test [74]. This test is also sometimes referred to as a model misspecification test. For example, in a panel data study, the test can be used to decide between two or more regression models, for example, the fixed effects model and the random effects model. This test has gained the trust and use of many researchers because of the simplicity of decision-making while interpreting its results; it is straightforward.

Decision Criteria:MG or PMG?Perform Hausman (1978) test
**H0**: MG and PMG are not significantly different, but PMG more efficient,**H1**: The null hypothesis is not trueDecision 1: Reject **H_0_** if *p*-value is less than 0.05, use MGDecision 2: Accept **H_0_** if *p*-value is greater than 0.05, use PMG

Please note that these decision criteria are basically the same if the comparison is to be PMG estimators and DFE or MG estimators and DFE.

## 3. Results

This section of the study presented the results and interpretation of the outlined estimation techniques using the materials, also captured in the last section.

### 3.1. Test of Collinearity

This test is necessary to ensure the model avoids multicollinearity or a collinearity problem. This simply means one variable can explain variation in the other and therefore should not be on the right side of equation [75].

There is no specific set of rules in modern econometrical analysis since the departure away from the initial definition of “perfect” or “exact” values. In Table 4 all the independent regressors are not linearly dependent on one another. The relationship with the highest correlation in the model is LnHxp (health expenditure) to LnLexp (life expectancy), which stands at 0.5175, absolute value. This is closely followed by the LnYpa (GDP per capita) to LnLexp (life expectancy) relationship, which stands at 0.3233. There was no correlation with substantive and significant value that could sum up to a collinear relationship. There were neither perfect nor exact linear relationships among the regressors in the model. The high values of correlation of 0.8407 and 0.7632 occurred between the dependent variable and LnHxp (health expenditure) to LnLexp (life expectancy) respectively. This is allowed and in accordance with literature. Therefore, the model passes either of collinearity or multicollinearity problems. It became saved from this point to continue to use the variables for the estimation of the model as earlier specified.

### 3.2. Descriptive Statistics Summary

Table 5 below shows the descriptive characteristics of each and every variable used in the model across all the panels. The statistics are panel-specific and not country-specific. The descriptive statistics generated are averages, standard deviation, minimum and maximum values.

### 3.3. Requisite Pre-Estimation Tests

The model was first set up in panel form, because there are a number of cross-sectional units involved; ten in all. The formal test of stationarity was adopted. The second-generation panel unit roots tests by Maddala and Wu [72] and IPS [76] were used. For the unit-roots confirmatory exercise, both the M and W ADF and IPS were conducted between at lags (0) to lag (3). Meanwhile, the decision rule for panel ARDL is that the variables can be integrated of order one, I (0) or I (1) or both, not I (2). The dependent variable must not be integrated of I (0). However, the lags of some of the variables can be distributed across I (0) and I (1). That is, at level and at first difference. Both test results are presented and interpreted below on Table 5, Table 6, Table 7 and Table 8 in Section 4.1.

#### Hausman (1978) Test

The proposed test of validity was conducted to determine the most suitable among the Pooled Mean Group (PMG), Mean Group (MG), and the Dynamic Fixed Effects (DFE) estimators. The Hausman (1978) test was conducted to ascertain level of statistical superiority of the estimators in this panel study. See result below in Table 6:

### 3.4. Panel ARDL Results: PMGE Long-Run and Group Short-Run Cointegration Estimates

In Table 7 the long-run estimates of the model were presented; attention to the first half of the table, which captured the long-run statistics of the PMGE. Cointegration was inferred from the results of the Pooled Mean Group (PMG) regression as estimated since long run homogeneity is assumed. In addition, the PMG estimator assumed that the short-run coefficients and the error correction terms are not the same across each of the countries, which is its heterogeneity property. The upper part of the table captured the long-run coefficients while the lower part presented the group short-run coefficients of the variables and the error correction term, that shows whether or not all the variables have a combined long-run cointegration.

### 3.5. Panel ARDL Results: PMG Heterogeneous Short-Run Cointegration Estimates

Having estimated the ARDL model using the PMG estimators for the group and for each of the ten countries, it was discovered that the null of long-run homogeneity was established, confirmed, and accepted. Therefore, the results of the individual error correction terms of the sampled countries are presented in Figure 6 below:

### 3.6. Institutional Assessment Index: CPIA

Unfortunately, the data for this variable has only been, warehoused from 2005 until present for five of the countries being reviewed, but bundled across sub-groups such as “resource rich sub-Saharan Africa countries (oil exporters)” and “resource rich sub-Saharan Africa countries (non-oil exporters)”, amongst others. It does not cover the entire period of this study. The good news is that the available data covers for more than a decade till present date. Therefore, the ability of these countries to create the needed strong institutional environment that supports diversified job creation by leveraging crude oil revenue to eventually attain sustainable development will be accessed from 2005 to 2017 across these two groups. The results are presented in the stacked columns charts below:

## 4. Discussion

### 4.1. Unit Roots Test

Table 8 and Table 9 show the results of the Maddala and Wu [72], and IPS tests unit root specifications at level for all the variables were presented. The results indicated that, under the augmented Dickey–Fuller assumptions, stationarity tests were conducted with constant/intercept (without trend) and with constant and time trend (with trend) different at level. Not all the variables satisfied the condition for cointegration test when tested at level. Therefore, the results were not enough to fulfil the condition for a panel ARDL model estimation. However, in Table 10 and Table 11, after taking the first difference of all the variables, it fluctuated around the zero mean; all from lag (0) to lag (3). This means they are all stationary, but their lags are distributed, because some of the variables would not need differencing again depending on the final optimal lag selected. This is a strong basis for using the panel ARDL model. Therefore, the variables were integrated of order I (1).

Null for MW and IPS tests: series is I(1).

MW test assumes cross-section independence.

IPS test assumes cross-section dependence is in form of a single unobserved common factor.

### 4.2. Hausman (1978) Test

In Table 6, the *p*-value of the Hausman (1978) test was 0.1392, which was more than 5% traditional critical level. Therefore, under the null hypothesis of homogeneity (that showed that PMG was the more efficient estimator) the alternative hypothesis was rejected. This simply means that MG is an unacceptable estimator for this model, given this result. Therefore, the model statistically supports PMG estimators for the estimation of the panel regression. Note should be taken that the result of the DFE vs. MG and DFE vs. PMG Hausman (1978) tests conducted showed that the chi-squared values in both instances were below zero and therefore explosive on the model. This further confirmed that the most appropriate of all the estimators’ models for the panel ARDL investigation is the Pooled Mean Group (PMG) estimator model.

### 4.3. PMGE Long-Run and Group Short-Run Cointegration Estimates

From Table 7 above, the coefficient of oil revenue (Ln_Oil_rent) showed a negative sign and a statistically insignificant long-run relationship with sustainable development (Ln_HDI). The *p*-value is 0.247 and the coefficient estimated is −0.003626. This is consistent with the theory of Dutch Disease that assumed a negative relationship between natural resource revenue and sustainable development in the long-run, provided no revenue diversification is done, leadership continues to be wasteful by misappropriating state funds, with attendant weak export and increasing purchase of military equipment to fight insurgencies [5,19,21,36]. Some of these studies categorically found out that oil revenue in African countries will continue to show less and less significance to regional and individual national development in the short-run, and no significance in the long-run, given the current level revenue misappropriation in these countries [19,55,77,78,79]. This current research has confirmed in a panel study what all these authors had attempted investigating in a micro or country-specific study. Hence, no long-run relationship between oil revenue and HDI, which measures sustainable development. However, the impact of this coefficient will be less than 1%, judging by the magnitude of the coefficient.

It was almost certain that GDP should have a positive long-run relationship with sustainable development. However, this result indicated a negative and significant relationship with HDI in the long run. this will not be a material increase. This is because, ceteris paribus, a 1% change in the value of GDP will only negatively impact sustainable development by approximately 0.24%. This is another indication that over-dependence on crude oil revenue in the long run will be grossly irrelevant to sustainable development [27,51,55,80,81,82]. GDP will become as nominal as it can be if not invested in human development drivers in the short run. This was in support of the review presented earlier. As mentioned earlier, experts have suggested diversification and investment in a linking sector [83].

In addition, government health expenditure, Lnhxp, will have a positive relationship with HDI, but the result is statistically insignificance at 0.179 *p*-value. The coefficient of contribution will increase HDI by 0.17% for every 1% change in the government health expenditure. It is believed that, considering the rate of neglect on health matters in the third world nations, outflow of health tourists among the haves and have-nots will be high, as it is growing already. Many will take to crowdfunding to raise health funds that will be spent overseas. Therefore, both the demand-side health expenditure and supply-side health expenditure will create an outflow or leakage rather than contribute to sustainable development [75,84,85,86]. Meanwhile, other estimates of HDI, gross national income per capita (Lnypa) and life expectancy (Lnlexp) indicated that they have a positive and significant long-run relationship with HDI at 4% and 129.5% contribution at every percentage change in value respectively. Their respective *p*-values are 0.0000; at 1% critical value. These values are in consonance with the a priori expectations and theories. It is expected that as technologies advance in these sampled countries, many people will seek alternate livelihood strategies, having been armed with increased knowledge on basic hygiene and security, to live longer and participate in an imposed inclusive economy [26,75,87,88,89].

Finally, because the panel cointegration model in Table 10 above was a group estimation using the PMG estimator, the error correction term (ECT) provided some glimpse of hope. It generated a significant coefficient which represented the speed of adjustment of the model to the long-run equilibrium. The ECT *p*-value was 0.010. Therefore, there was an observed long-run relationship among the variables in the panel within the 1% level. In addition, the coefficient of the ECT showed that any deviation from the long-run equilibrium is corrected at −0.5417982 adjustment speed. Meanwhile, note should be taken that the long-run coefficients of the PMG estimator is the same or homogenous for all the countries sampled, but their respective short-run coefficients will differ, including the ECTs. This will be captured when the full PMG estimator coefficient is presented to indicate heterogeneity of the ECT variances and the short-run coefficients. This is a key assumption of the PMG estimators. Therefore, there is cointegration GDP, GNI per capita, and life expectancy are all cointegrated with HDI in the long run, while oil revenue and health expenditure are not.

### 4.4. PMG Heterogeneous Short-Run Cointegration Estimates

Table 12 and Figure 4 show the presentation of the short-run error correction terms for each of the countries and a plot of the count of significance. The table contains the coefficients, which represent the speed of adjustment to equilibrium and the *p*-values that suggest statistical significance or otherwise. The graph shows the count of three countries with insignificant ECT in the short run. These countries are Angola and Equatorial Guinea. Others maintained significant results. Meanwhile, note that this presentation was necessary because the study is a heterogeneous dynamic panel data model, otherwise called panel ARDL. ARDL assumes heterogeneity and homogeneity of short-run and long-run coefficients respectively. Therefore, the individual error correction terms (ECT) was estimated to understand the adjustment speed for each of the countries, should there be deviation from the long-run equilibrium.

### 4.5. Oil Revenue: Panel ARDL Estimates

This section singled out the coefficients of the model with respect to oil revenue across the panel. It was expected to show more clearly the cohesion of arguments from the literature review and the findings of the results presented in this chapter.

Table 12 above showed the outcome of the decision rule, having compared the *p*-values of the heterogeneous PMG estimator’s short-run coefficients of oil revenue by the reviewed countries. The results showed that five of the countries (Algeria, Congo, Egypt, Sudan, and Nigeria) had *p*-values within the 5% band, though the short-run coefficients were negative for oil revenue.

Table 13 and Figure 7 presented the significance of oil revenue across each of the countries. The PMG panel ARDL model produced a heterogeneous result that made it possible for the performance of each country to be observed in the short run. The table shows the negative coefficient of crude oil revenue in Angola, which might be a reflection of things to befall all the oil giants in Africa, as indicated from the long-run combined negative relationship of oil revenue with sustainable development. Therefore, Angola’s 1% change in oil revenue will contribute −1.4% to sustainable development. This is evident in the various misplaced programs created by the Angolan government. One of such was the building of a multi-million-dollar city that has turned to a ghost city because it was a wrong social investment [89,90]. Therefore, the relationship is not statistically significant at 0.615 *p*-value. Other countries contribute positively in the short-run, but only Equatorial Guinea produced a significant *p*-value at 0.000 with 5.3% coefficient of contribution to HDI. At the top on the list of these countries is Chad with a 6.3% coefficient of contribution, while the rest contribute between 1.4% and 0.2% maximum and minimum, respectively. These results show that the way oil revenue is used in these countries is not projected at advancing sustainable development.

### 4.6. Causality Tests

One of the characteristics of the PMG estimators is its informativeness. The coefficients interact so much that a number of inferences can be drawn from them. In the light of this, causality will be inferred from the statistics of the PMG estimators earlier produced and presented.

#### 4.6.1. Long-Run Causality

Long-run causal effects are inferred when the long-run coefficients of the PMG estimators are statistically significant. The rules are:Long-run coefficients for long-run causalityShort-run coefficients for short-run causalityBoth period coefficients for strong causality

Table 14 shows long-run causality results among the variables and the dependent variables. However, the estimator clearly shows causality and the causal periods. While the null hypotheses of no causality were accepted for oil revenue and health expenditure, it was found that long-run causal relationships existed between Ln_GDP and Ln_HDI, GNI per capita and Ln_HDI, and life expectancy and HDI. In addition, a joint long-run relationship was observed from all the regressors to the dependent variable, HDI.

#### 4.6.2. Short-Run Causality

Similarly, short-run causal effects are inferred when the short-run coefficients of the PMG estimators are statistically significant. The rules are:Short-run coefficients for short-run causalityError correction term for joint causalityBoth error correction term and short-run coefficients for strong causality

Table 15 shows joint causal relationships from the explanatory variables to the dependent variable in all sampled countries except Angola and Equatorial Guinea. This was clear from the column where H_0_ was rejected.

Table 16 below captured the various causality hypotheses necessary to understand how oil revenue reacted in the long-run and short-run with the dependent variable. The null of no causality was accepted across all the samples except for Algeria, Equatorial Guinea, and Gabon. These varying results are consistent with findings of country-level empirical studies and some cross-country/panel such as [32,39,91,92,93].

### 4.7. Institutional Assessment Index: CPIA

The stacked columns in Figure 8 show the ratings of resource-rich crude oil-exporting and non-crude oil-exporting African countries. The former is dominated by the top-ten oil-exporting countries, with over 74% revenue contributions to GDP on the average and over 95% resource revenue contributions on average as well. The oil exporters are captured on the red-coloured stacked columns while the non-oil exporters are captured on the blue-coloured stack columns. CPIA on building human resources evaluates the policies, on the public frontier’s service delivery, that encourage or hinder access to quality health and education services, cum deterrence and treatment of HIV/AIDS, tuberculosis, and malaria”, which basically define HDI [60]. In 2015, the non-oil exporters are seen to be better with 3.25 against 2.92 of the oil exporters. Though the rating of non-oil exporters grew slightly in subsequent years, recording ranking that ranged between 3.29 and 3.50 in 2013 that of the oil exporters dropped to between 2.83 to 2.93 in the same year. However, both groups moved within the same rank in 2016, but the non-oil exporters rank value stood at 3.58 as against 3.08 of the oil exporters (There was no value captured for 2017 as at the time of the research). This simply suggests that a need for proper investigation to know why a wealthier group of oil-exporting African countries would be more fragile to deal with than less wealthy and non-oil exporting African countries. It further confirms that a lot of improprieties that mar processes of operation of governance and resource usage among the crude oil-rich countries.

Similarly, Figure 9 is a set of stacked columns that show the ratings of resource-rich crude oil exporters and non-crude oil exporters African countries in terms of CPIA by the World Bank. The legends arrangements are as earlier described except that the measures are different. The CPIA here are meant to measure the degree of fragility of a country with respect to how its policies prevent or entrench transparency, accountability, and corruption, here captured from 2005 to 2016. Clearly, the non-oil exporting countries promised a better rating than the oil-exporting African countries. This is because, though both groups are within the same band of rating, but the non-oil African countries have always been closer to the next band, 3.0, since 2005. This inferred that there is a greater degree of lack of transparency and accountability accompanied by corruption in the public and private sectors of the economies of the oil-exporting countries than the non-oil-exporting countries. This confirms why a country such as Nigeria, arguably the highest crude oil-producing country in Africa and 6th in the world, has been consistently classified as among the top ten most corrupt nations of the world within about the same period of data coverage [59,94,95,96].

### 4.8. Diagnostic Checks

Figure 10 below showed a bell-shaped distribution of the model residuals, with the residuals captured on the X-axis and the data set density on the Y-axis [66]. Meanwhile, Figure 11 is an informal presentation of the plot of the squared residuals to see if a definite pattern is observable from the image. From the plot, it is clear that no definite pattern could be observed. Therefore, this confirms that the model is homoscedastic.

## 5. Conclusions

The study investigated the relationship between sustainable development and oil revenue in ten crude oil-producing African counties. The selection criteria were crude oil revenue and production capacity as presented in the introductory section of the study. The problem statement of the study highlighted issues such as overdependence on oil revenue, mono-economic policy and non-diversified crude oil proceeds, and low human development index (HDI). The main objective of the study was to find out if this overdependence on oil revenue would contribute positively and significantly to sustainable development, measured by the Human Development Index (HDI) across the selected countries.

In the empirical and theoretical literature review, an outlook of the selected African crude oil-producing giants was presented within twenty-six years, 1992 to 2017. This was done across related articles and relevant theories. There, relevant macroeconomic indices were evaluated to complement the analyses around HDI and oil revenue. It was observed that oil revenue was not necessarily responsible for the level of sustainable development, measured by HDI, attained in these countries, but the use to which they put the revenue. Oil revenue grew, but HDI responded at a much slower rate in comparison. The findings of the study suggested that oil revenue should be used to create diversified sources of revenue and invested in core products and services that contribute directly to sustainable development. These include health, education, agriculture, and other life-improving basic needs for human welfare. The various articles reviewed gave varying findings on the relationship between HDI and crude oil revenue (COR). All of them advocated for alternatives to natural resources’ revenue because of possible depletion, revenue volatility, and other ills that are attributable to institutional porosity, especially within the public systems where fiscal decisions are made and executed.

Panel ARDL was presented as the estimation technique because the study was dynamic heterogeneous panel research. Consequently, the empirical results presented showed that crude oil revenue (COR) contributed positively and significantly to HDI in some of the countries, why some recorded positive but insignificant relationship between both variables in the short run. However, one country in particular, Angola, recorded a negative and insignificant oil revenue-HDI short-run relationship. These results are in consonance with the framework reviewed. They opined that resource revenue will lead to growth if the proceeds are used to create artificial capital that will in tune create the required ripple effects for sustainable development. Countries with positive but insignificant relationship between crude oil revenue and HDI might have justified the conclusions of the theory of unbalanced growth (TUG). TUG concluded that booming sectors can only have effect if their proceeds are invested in sectors that have linkages across all sectors; such sectors with higher velocity to promote sustainable development. Meanwhile, in the long-run, given the current condition of overdependence on crude oil revenue, poor revenue and investment management, porous leadership, and other institutional factors, crude oil revenue will have a joint negative impact on HDI. However, the long-run error correction term (ECT) produced the right coefficient and statistical significance needed as the speed of adjustment to the required equilibrium for all variables combined.

The outcomes of this study have raised various policy issues and recommendations. The empirical results have shown that a short-run positive relationship existed between sustainable development and crude oil revenue in all the countries, but the long-run negative output can be avoided with the respite provided by the speed of adjustment given by the ECT coefficient in the long-run. This suggested that crude oil revenue is not bad in itself. It is now expedient for utmost fiscal and monetary priority and attention to be directed at using the proceeds of crude oil revenue optimally and on the back of a well-organized economic institution, devoid of corruption and open to accountability, transparency, and inclusiveness. That said, specific suggestions such as economic diversification, production, and consumption of local content in sectors that possess forward and backward linkages to other sectors, such as agriculture, focus on increasing health, education, life expectancy and income per capita will help change the tide of things. Finally, supply-side policies could be used to improve the competitiveness of domestic industries. Supply-side policies are usually aimed at increasing aggregate supply by enhancing the productive capacities and quality of all factors of production. Such policies that will encourage fiscal flexibility to expand local investment and drive down the natural rate of unemployment. Supply-side policies are expected to aid freer movement of resources across the economy. Broadly, supply-side policies are divided into regulatory, tax, and monetary policies. Suggestions were made that a fully digitalized economy will discourage corruption, promote accountability, transparency, continuity, and development stability.

The study investigated the relationship between sustainable development and oil revenue in ten crude oil-producing African countries for the period 1992 to 2017. The choice of countries as well as the chosen period was determined by the availability of data and research funding. The findings have shown that in the long run, the oil revenue will not be significant to growing sustainable development, but rather contribute negatively if the former’s use is not properly diversified and deployed with the support of efficient and effective public institutions. Therefore, future research studies can be conducted in this area.

## Figures and Tables

**Figure 1 ijerph-17-06799-f001:**
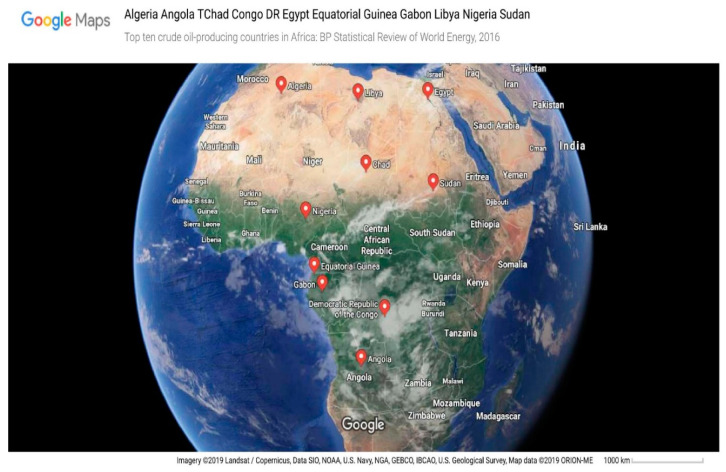
Maps of top ten crude oil-producing African countries. Source: Google Map, (2019).

**Figure 2 ijerph-17-06799-f002:**
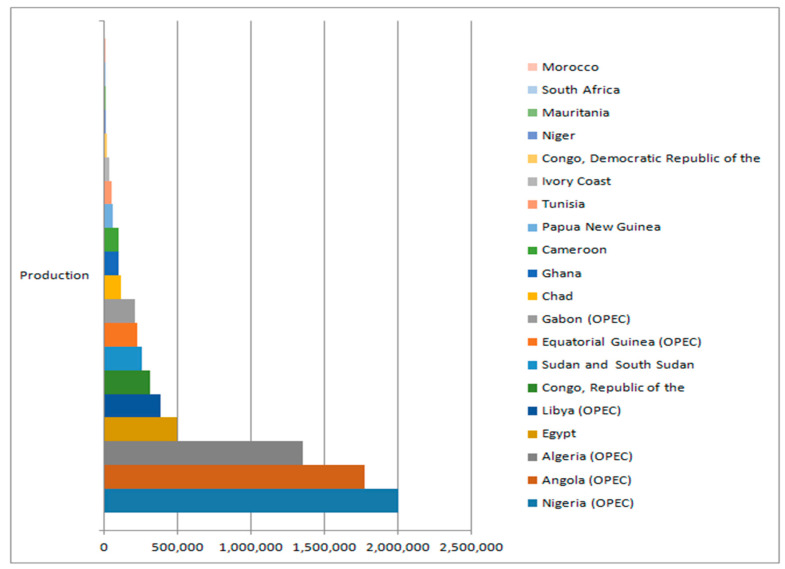
Oil distribution by daily production per million barrels. Source: World Bank data group.

**Figure 3 ijerph-17-06799-f003:**
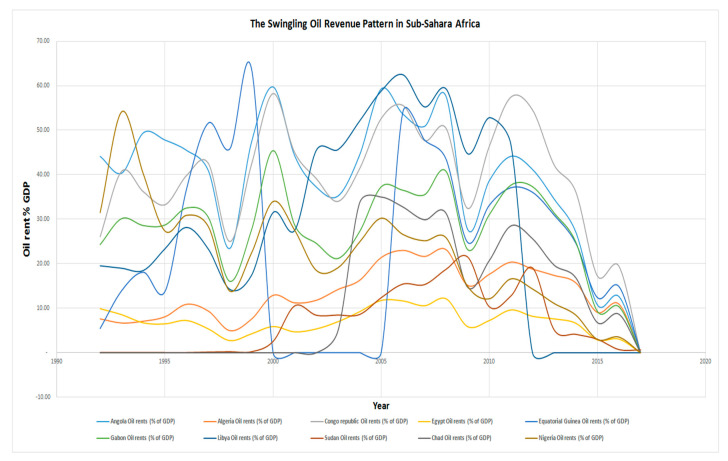
Oil rent in top ten crude oil-producing African countries. Source: World Bank data group.

**Figure 4 ijerph-17-06799-f004:**
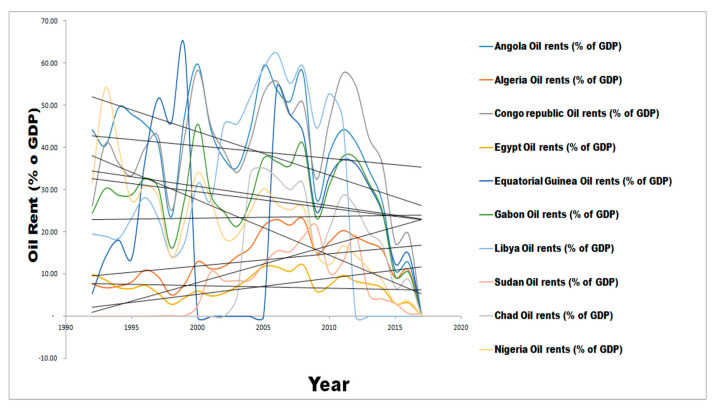
Oil rent in top ten crude oil-producing African countries - with trend lines. Source: World Bank data group.

**Figure 5 ijerph-17-06799-f005:**
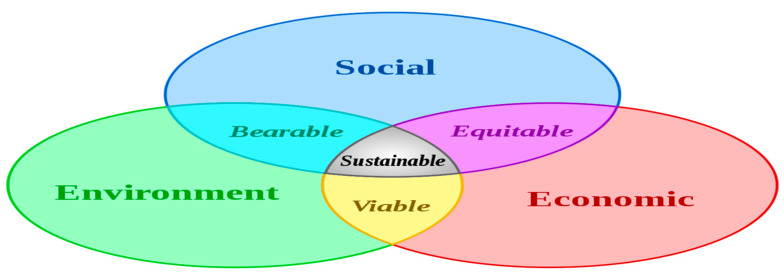
Model of “Sustainable Development” [8]. Source: Richard Howarth [8].

**Figure 6 ijerph-17-06799-f006:**
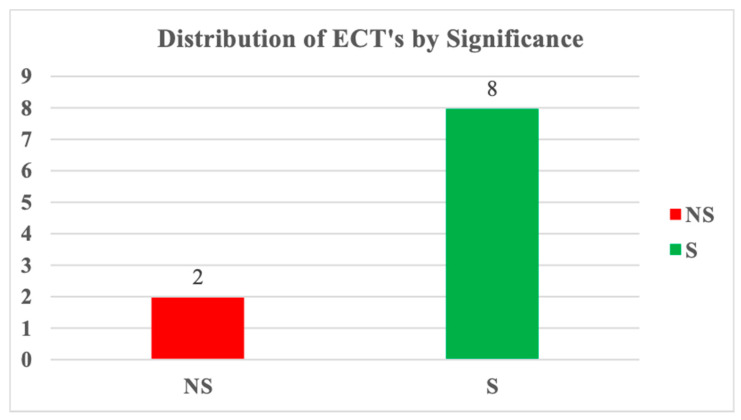
Short-run ECT’s. Source: World Bank data group. NS—No significant, S—Significant.

**Figure 7 ijerph-17-06799-f007:**
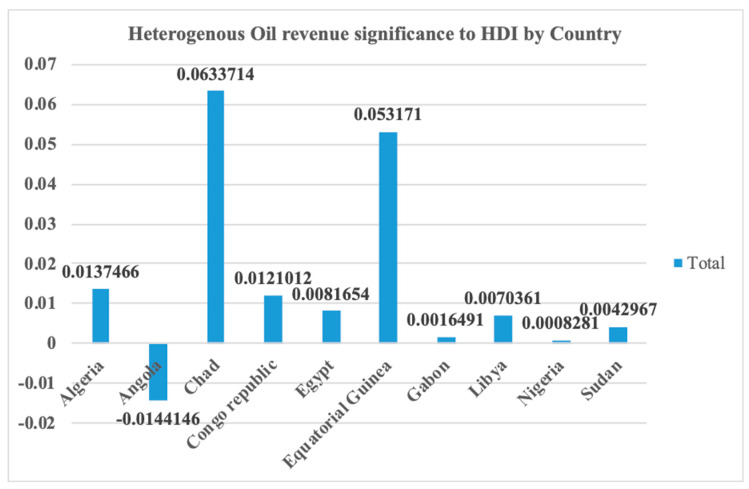
Oil revenue’s significance to sustainable development by country. Source: World Bank group data, (2019).

**Figure 8 ijerph-17-06799-f008:**
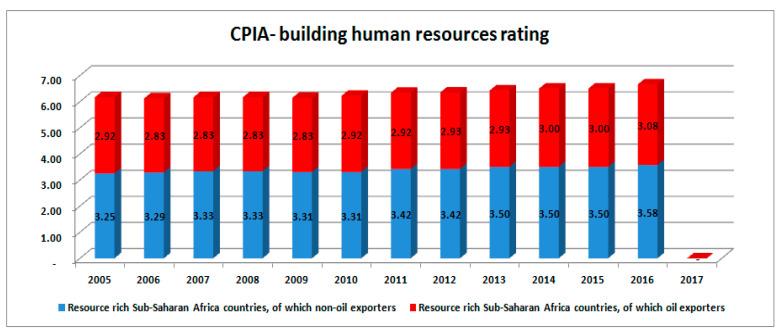
Country policy and institutional assessment: building human resources rating.

**Figure 9 ijerph-17-06799-f009:**
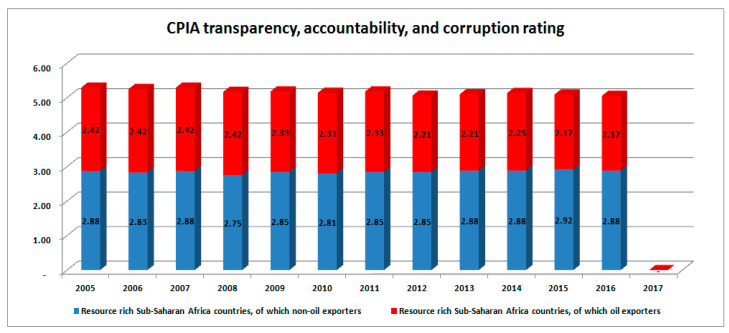
CPIA transparency, accountability, and corruption in the public sector rating. Source: World Bank data group.

**Figure 10 ijerph-17-06799-f010:**
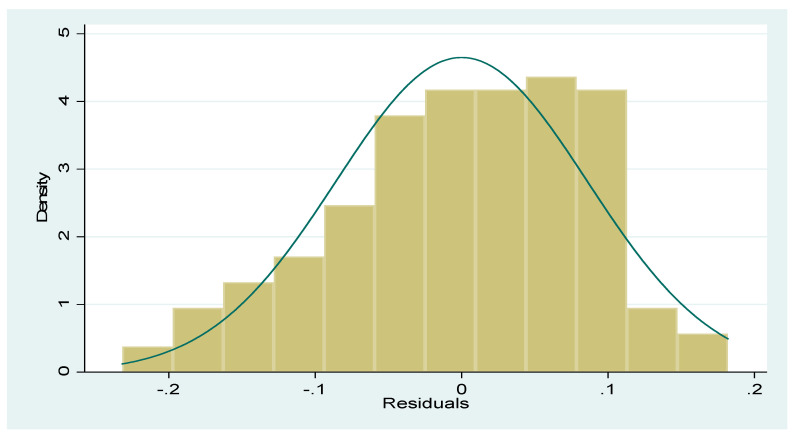
Residual normality plot—bell shaped histogram. Source: World Bank data group.

**Figure 11 ijerph-17-06799-f011:**
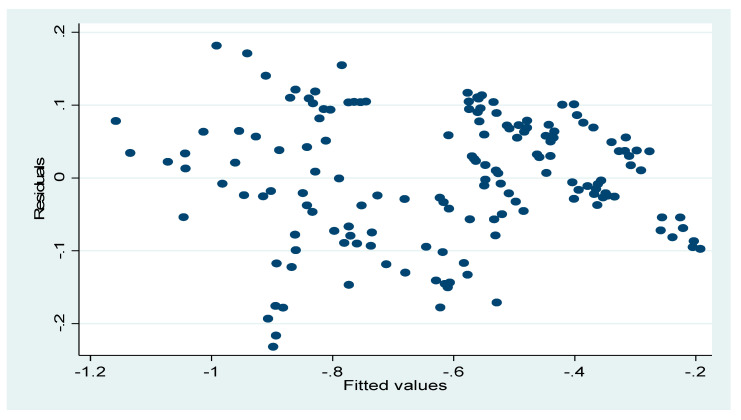
Residual-fitted values plot. Source: World Bank data group.

**Table 1 ijerph-17-06799-t001:** World Bank HDI and crude oil production ranking.

HDI Rank	Country	HDI 2015	HDI Cadre	Oil Prod. *(mbpd)*	Membership
83	Algeria	0.745	High	1,348,361	OPEC
102	Libya	0.716	High	384,686	OPEC
111	Egypt	0.698	Medium	494,325	NON-OPEC
109	Gabon	0.697	Medium	210,820	OPEC
135	Equatorial Guinea	0.592	Medium	227,000	OPEC
152	Nigeria	0.527	Low	1,999,885	OPEC
150	Angola	0.516	Low	1,348,361	OPEC
167	Sudan	0.493	Low	255,000	NON-OPEC
176	Congo Republic	0.418	Low	308,363	NON-OPEC
186	Chad	0.396	Low	110,156	NON-OPEC

Source: World Development Index. (mbpd: Oil production is described in *million barrels per day (mbpd)*).

**Table 2 ijerph-17-06799-t002:** Elements of UN’s Human Development Index (HDI).

Education	Health	Income
Average years of schooling Expected years of schooling	Birth life expectancy	Gross national income per capita

Source: UNDP, (2018).

**Table 3 ijerph-17-06799-t003:** Crude oil net exports by country—the world’s top 15 in 2018.

Country	Crude Oil Net Export 2018
1. Saudi Arabia:	USD$182.5 billion (net export surplus down−27.1% since 2014)
2. Russia	$129 billion (down−16%)
3. Iraq	$91.7 billion (up 9%)
4. United Arab Emirates	$55.9 billion (down−22.4%)
5. Canada	$52 billion (down−22%)
6. Kuwait	$51.7 billion (down−25.4%)
7. Iran	$50.8 billion (up 30.5%)
8. Nigeria	$43.6 billion (down−39.8%)
9. Kazakhstan	$37.8 billion (down−29.3%)
10. Angola	$36.5 billion (down−35.3%)
11. Norway	$31.6 billion (down−30.4%)
12. Libya	$26.7 billion (up 69.4%)
13. Mexico	$26.4 billion (down−25.9%)
14. Venezuela	$26.4 billion (down−50.9%)
15. Oman	$22.5 billion (down−35.5%)

Source: Investopedia and British Petroleum.

**Table 4 ijerph-17-06799-t004:** Correlation analysis.

Variables	Ln_Hdi	Ln_Oil_Rent	Lnypa	Lnlexp	Ln_Gdp	Lnhxp
ln_hdi	1.0000					
ln_oil_rent	−0.1017	1.0000				
Lnypa	−0.0229	0.2772	1.0000			
Lnlexp	0.7632	−0.2072	−0.3233	1.0000		
ln_gdp	0.0638	−0.0467	−0.1050	−0.0473	1.0000	
Lnhxp	0.8407	−0.0591	−0.0237	0.5175	0.1058	1.0000

Source: World Bank data group.

**Table 5 ijerph-17-06799-t005:** Descriptive statistics of the observed variables.

Variable	Obs	Mean	Std. Dev.	Min	Max
ln_hdi	260	−0.5360769	0.3194807	−1.21	0
ln_oil_rent	260	2.348885	1.402715	−2.18	4.16
ln_gdp	260	20.36323	5.139646	0	27.07
lnhxp	260	2.016269	1.986721	−1.69	5.64
lnypa	260	11.26362	3.395723	−8.11	16.22
lnlexp	260	4.072923	0.1528188	3.81	4.34

Source: World Bank data group.

**Table 6 ijerph-17-06799-t006:** Hausman (1978) Test.

	Coefficients			
	(b)	(B)	(b-B)	Sqrt (diag(V_b-V_B))
	**Mg**	**Pmg**	**Difference**	**S.E.**
ln_Oil_Rent	0.0134381	−0.0036264	0.0170645	0.2337038
ln_Gdp	−0.1172880	−0.0023694	−0.1149186	0.1242981
Lnhxp	0.1742662	0.0017171	0.1725490	0.2324977
Lnypa	−0.2235028	0.0399573	−0.2634600	0.3257325
Lnlexp	0.2532273	1.2958560	−1.0426290	2.5626270
b = consistent under Ho and Ha; obtained from xtpmg				
B = inconsistent under Ha, efficient under Ho; obtained from xtpmg				
Test: Ho: difference in coefficients not systematic				
	chi2(5) = (b-B)’[(V_b-V_B)^(−1)](b-B)			
		=	8.33	
	Prob > chi2	=	0.1392	

Source: World Bank data group.

**Table 7 ijerph-17-06799-t007:** Long run coefficients of the PMG estimators.

Pooled Mean Group Regression						
D.Ln_HDI	Coef.	Std. Err.	z	*p* > z	[95% Conf. Interval]	
ECT						
ln_Oil_Rent	−0.003626	0.003134	−1.16	0.247	−0.0097695	0.0025167
ln_Gdp	−0.002369	0.000528	−4.49	0.000	−0.0034039	−0.001335
Lnhxp	0.0017171	0.001279	1.34	0.179	−0.0007898	0.0042241
Lnypa	0.0399573	0.003119	12.81	0.000	0.0338444	0.0460701
Lnlexp	1.295856	0.089509	14.48	0.000	1.120422	1.47129
SR						
ECT	−0.541798	0.211398	−2.56	0.010	−0.9561296	−0.127467
ln_Oil_Rent						
D1.	0.0122458	0.008182	1.5	0.134	0.0037902	0.0282818
ln_Gdp						
D1.	0.0020286	0.001429	1.42	0.156	0.0007713	0.0048285
Lnhxp						
D1.	−0.027886	0.025276	−1.1	0.270	0.0774253	0.0216533
Lnypa						
D1.	0.0262084	0.029854	0.88	0.380	0.0323051	0.084722
Lnlexp						
D1.	4.143945	2.323807	−1.78	0.075	8.698522	0.410633
_cons	3.409108	1.339973	−2.54	0.011	6.035408	0.7828085

Source: World Bank data group.

**Table 8 ijerph-17-06799-t008:** Maddala and Wu, (1999) ADF unit roots test results.

Level		Specification without Trend		Specification with Trend	
Variable	Lags	Chi Square	*p*-Value	Chi Square	*p*-Value
Ln_HDI	0	23.8202	0.2503	47.4228	**0.0005**
Ln_HDI	1	16.768	0.6680	19.9911	0.4585
Ln_Oil_rent	0	98.4007	**0.0000**	73.3724	**0.0000**
Ln_Oil_rent	1	49.858	**0.0002**	25.6001	0.1794
Ln_GDP	0	186.4398	**0.0000**	195.2186	**0.0000**
Ln_GDP	1	87.7425	**0.0000**	86.9921	**0.0000**
Lnhxp	0	18.3181	0.5665	47.9663	**0.0004**
Lnhxp	1	21.83	0.3498	30.269	0.0656
LnYpa	0	94.1657	**0.0000**	26.4474	0.1515
LnYpa	1	55.9286	**0.0000**	32.1493	**0.0417**
LnLexp	0	7.6517	0.9939	32.795	**0.0355**
LnLexp	1	10.8991	0.9488	14.6928	0.7937

Source: World Bank data group. All bold *p*-Values within 0.05 show significance where H_0_ is rejected. That is, the referenced variables are stationary.

**Table 9 ijerph-17-06799-t009:** Maddala and Wu, (1999) ADF unit roots test results at first difference.

First Difference		Specification without Trend		Specification with Trend	
Variable	Lags	Chi Square	*p*-Value	Chi Square	*p*-Value
D.Ln_HDI	0	316.9644	**0.0000**	284.9562	**0.0000**
D.Ln_HDI	1	110.274	**0.0000**	88.4604	**0.0000**
D.Ln_Oil_rent	0	411.9189	**0.0000**	387.5273	**0.0000**
D.Ln_Oil_rent	1	129.3378	**0.0000**	120.9628	**0.0000**
D.Ln_GDP	0	524.788	**0.0000**	447.9156	**0.0000**
D.Ln_GDP	1	280.9217	**0.0000**	233.5367	**0.0000**
D.Lnhxp	0	87.9795	**0.0000**	68.7034	**0.0000**
D.Lnhxp	1	92.08	**0.0000**	113.1743	**0.0000**
D.LnYpa	0	139.5193	**0.0000**	155.6993	**0.0000**
D.LnYpa	1	71.3424	**0.0000**	72.8166	**0.0000**
D.LnLexp	0	234.343	**0.0000**	232.1674	**0.0000**
D.LnLexp	1	96.9863	**0.0000**	121.2085	**0.0000**

All bold *p*-Values within 0.05 show significance where H_0_ is rejected. That is, the referenced variables are stationary. Source: Word Bank data group.

**Table 10 ijerph-17-06799-t010:** Peseran, (2007) Panel Unit Roots test (IPS).

Level		Specification without Trend			Specification without Trend		
Variable	Lags	Zt-Bar	*p*-Value	w-t-Bar	Zt-Bar	*p*-Value	w-t-Bar
Ln_HDI	0		0.7086	0.5492		0.0327	−1.8423
Ln_HDI	1		0.9803	2.0600		0.7638	0.7187
Ln_Oil_rent	0		**0.0000**	−6.3685		**0.0000**	−4.6169
Ln_Oil_rent	1		**0.0008**	−3.1717		0.2776	−0.5900
Ln_GDP	0		**0.0000**	−10.7884		**0.0000**	−11.3298
Ln_GDP	1		**0.0000**	−5.8243		**0.0000**	−5.8955
Lnhxp	0		0.4021	−0.2478		0.9019	1.2922
Lnhxp	1		0.2241	−0.7586		0.8800	1.1749
LnYpa	0		**0.0001**	−3.7687		0.9691	1.8672
LnYpa	1		**0.0137**	−2.2057		0.9080	1.3284
LnLexp	0		1.0000	6.5234		0.1664	−0.9687
LnLexp	1		0.9987	3.0161		0.8527	1.0482

All bold *p*-Values within 0.05 show significance where H_0_ is rejected. That is, the referenced variables are stationary.

**Table 11 ijerph-17-06799-t011:** Peaseran, (2007) Panel Unit Roots test (IPS).

First Difference		Specification without Trend			Specification without Trend		
Variable	Lags	Zt-Bar	*p*-Value	w-t-Bar	Zt-Bar	*p*-Value	w-t-Bar
D.Ln_HDI	0		**0.0000**	−15.2053		**0.0000**	−14.3681
D.Ln_HDI	1		**0.0000**	−0.5233		**0.0000**	−5.3876
D.Ln_Oil_rent	0		**0.0000**	−18.654		**0.0000**	−18.0981
D.Ln_Oil_rent	1		**0.0000**	−7.8958		**0.0000**	−7.2173
D.Ln_GDP	0		**0.0000**	−22.527		**0.0000**	−20.9120
D.Ln_GDP	1		**0.0000**	−13.865		**0.0000**	−12.2606
D.Lnhxp	0		**0.0000**	−5.3624		**0.0001**	−3.8454
D.Lnhxp	1		**0.0000**	−4.1343		**0.0000**	−4.0031
D.LnYpa	0		**0.0000**	−8.3336		**0.0000**	−9.1850
D.LnYpa	1		**0.0000**	−4.8857		**0.0000**	−4.8438
D.LnLexp	0		**0.0000**	−11.23		**0.0000**	−11.685
D.LnLexp	1		**0.0000**	−5.4079		**0.0000**	−5.2434

All bold *p*-Values within 0.05 show significance where H_0_ is rejected. That is, the referenced variables are stationary. Source: World Bank data group.

**Table 12 ijerph-17-06799-t012:** Short-run ECTs, sampled countries.

Country	Coefficient	*p*-Value
Angola	−0.1568364	0.153
Algeria	−0.6003596	0.001
Congo	−0.9785625	0.000
Egypt	−2.244512	0.000
Equatorial Guinea	−0.061304	0.150
Gabon	−0.0890633	0.020
Libya	−0.2241235	0.061
Sudan	−0.6560315	0.000
Chad	0.1735553	0.047
Nigeria	−0.2336334	0.009

ECT is the error correction term that specify the speed of adjustment of the insignificant variables to equilibrium in the long-run, ceteris paribus. Source: World Bank data group.

**Table 13 ijerph-17-06799-t013:** Panel ARDL oil revenue, short-run coefficients.

Country.	Coef.	Std. Err.	z	*p* > z	Decision
Angola	−1.4%	0.0286231	−0.5	0.615	NS
Algeria	1.4%	0.0081675	−1.68	0.092	S
Congo	1.2%	0.0076664	1.58	0.114	NS
Egypt	0.8%	0.0050601	1.61	0.107	NS
Equatorial Guinea	5.3%	0.0099405	5.35	0.000	S
Gabon	0.2%	0.0009845	1.68	0.094	S
Libya	0.7%	0.0223271	0.32	0.753	NS
Sudan	0.4%	0.0069608	0.62	0.537	NS
Chad	6.3%	0.0487136	1.3	0.193	NS
Nigeria	0.1%	0.0285054	0.03	0.977	NS

Where: NS = Not statistically significant; S = Statistically significant. Source: World Bank group data, (2019).

**Table 14 ijerph-17-06799-t014:** Long-run causality summary.

Long-Run Causality Summary			
Null Hypothesis	z	*p*-Value	Causality
There is no long-run causality between Oil Revenue and HDI	−1.16	0.247	Accept Ho
There is no long-run causality between GDP and HDI	−4.49	0.000	Reject Ho
There is no long-run causality between health expenditure and HDI	1.34	0.179	Accept Ho
There is no long-run causality between GNI per capita and HDI	12.81	0.000	Reject Ho
There is no long-run causality between life expectancy and HDI	14.48	0.000	Reject Ho
There is no joint long-run causality of the regressors on HDI in the long-run	−2.56	0.010	Reject Ho

Source: World Bank data group—Inferred from the PMG estimator’s results.

**Table 15 ijerph-17-06799-t015:** ECT joint causality summary.

Country	Null Hypothesis	*p*-Value	Causality
Angola	There is no joint causality between all the regressors and HDI	0.153	Accepts Ho
Algeria	There is no joint causality between all the regressors and HDI	0.001	Rejects Ho
Congo	There is no joint causality between all the regressors and HDI	0.000	Rejects Ho
Egypt	There is no joint causality between all the regressors and HDI	0.000	Rejects Ho
Equatorial Guinea	There is no joint causality between all the regressors and HDI	0.150	Accepts Ho
Gabon	There is no joint causality between all the regressors and HDI	0.020	Rejects Ho
Libya	There is no joint causality between all the regressors and HDI	0.061	Rejects Ho
Sudan	There is no joint causality between all the regressors and HDI	0.000	Rejects Ho
Chad	There is no joint causality between all the regressors and HDI	0.047	Rejects Ho
Nigeria	There is no joint causality between all the regressors and HDI	0.009	Rejects Ho

Source: World Bank data group.

**Table 16 ijerph-17-06799-t016:** Oil Revenue-HDI Causality Summary.

Country	Null Hypothesis	Z	*p*-Value	Causality
Angola	There is no short-run causality between oil revenue and HDI	−0.5	0.615	Accepts Ho
Algeria	There is no short-run causality between oil revenue and HDI	−1.68	0.092	Reject Ho
Congo	There is no short-run causality between oil revenue and HDI	1.58	0.114	Accepts Ho
Egypt	There is no short-run causality between oil revenue and HDI	1.61	0.107	Accept Ho
Equatorial Guinea	There is no short-run causality between oil revenue and HDI	5.35	0.000	Reject Ho
Gabon	There is no short-run causality between oil revenue and HDI	1.68	0.094	Reject Ho
Libya	There is no short-run causality between oil revenue and HDI	0.32	0.753	Accepts Ho
Sudan	There is no short-run causality between oil revenue and HDI	0.62	0.537	Accepts Ho
Chad	There is no short-run causality between oil revenue and HDI	1.3	0.193	Accepts Ho
Nigeria	There is no short-run causality between oil revenue and HDI	0.03	0.977	Accepts Ho

Source: World Bank data group.

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
