# Peer review of "Sustainable Development and Crude Oil Revenue: A Case of Selected Crude Oil-Producing African Countries"

_ijerph, 2020, doi:10.3390/ijerph17186799_

Round 1

Reviewer 1 Report

The author/s elaborate an article applied a quasi-experimental design to evaluate the resulting influence on the social empowerment of farmers and on consumer diets.
The structures and contents of the manuscript should be further reorganized and improved, respectively.
The introduction must provide an extensive overview of recent developments in this specific area that fall within the scope of the journal and not only a list of published studies.
The objectives of this study should be clearly stated to present the novelty of the study in the Introduction.
The originality and significance of the paper needs to be further clarified.
The methodology is explained more clearly.
The current results and discussion should be significantly extended with the possible reasons and practical application of the findings.
In the Conclusion, please discuss the implications of your research. Conclusion should be revised and go deeper, it would be more interesting if the authors focus more on the significance of their findings regarding the importance of the interrelationship between the obtained results and the real application.

Author Response

Dear Reviewer,

The comments are suggestions for author/s have been addressed in the attached response. All indicated improvement from the open reviews have also been addressed. Kindly see the uploaded documents for further details.

Thank you.

Ifeoluwa A. Ologunde,
Corresponding Author
ife.gold234@gmail.com 

Reviewer 2 Report

This is an interesting article from a theoretical point of view as well as from a practical perspective. Methods of research work are appropriate to the research purpose, and the aim of the paper has been fulfilled.

I recommend writing “1.2 Literature review” instead of “1.2 Literature reviews and relevant frameworks”

Instead of “Sustainable Development” by: Johann Dreo” I recommend writing “Figure 3.1: Model of “Sustainable Development”, to cite source with number between square bracket [0] and to include the full reference in the references list: 1.- Please always cite sources by using a number between square brackets [0], 2.- According to academic standards each reference cited in text must appear in the reference list, and each entry in the reference list must be cited in text, id est, cite what you use, use what you cite.

Page 8 must not be almost empty.

It is written in an academically appropriate style. It is easy to read, with logical sequencing of information, and it fulfils academic expectations. It has been conducted a very good statistical analysis process. This paper is well founded and comprehensible in its reasoning. Conclusion is well substantiated and comprehensible.

Author Response

Dear Reviewer,

Please see attached and below for tabulated point by point detailed responses on the suggestions and issues raised.

Reviewer 2

Summarized review point:

  1. Article is theoretically, empirically, and econometrically interesting Responses: Commendation acknowledged. Relevant Pages: NA
  2. I recommend writing “1.2 Literature review” instead of “1.2 Literature reviews and relevant frameworks; Responses: Correction made; Relevant Pages: See line 174, Page 7
  3. Figure 3.1: Model of “Sustainable Development”, to cite source with number between square bracket; Responses: This has been corrected on the scenario highlighted and across the document; Relevant Pages: See page 7 and other similar tracked changes in the latest version of the manuscript.
  4. Page 8 must not be almost empty; Responses: Re-organized accordingly Relevant Pages: See page 8
  5. Writing was academically appropriate in style, logical presentation, very good statistical analysis process; Responses: Commendation acknowledged. Relevant Pages: NA

Thank you.

Ifeoluwa A. Ologunde
Corresponding Author
ife.gold234@gmail.com

Reviewer 3 Report

The paper investigates the relationship between sustainable development and crude oil revenue in top ten oil-producing African countries in the period 1992-2017, using the Pooled Mean Group (PMG) estimators on panel autoregressive distributed lag model (ARDL). Human Development Index (HDI) is chosen as a proxy for sustainable development. No long-term relationship between oil revenue and sustainable development is found.

The topic this paper addresses is relevant, since oil-exporting countries could use oil revenues to promote human capital investment (education and health) in order to achieve sustainable development and improve social welfare. Thus, oil revenues could be viewed as a way to escape poverty for oil-exporting developing countries.

Below are some of my thoughts on the paper.

The results are far from being unexpected owing to the socioeconomic and political features of countries in the sample. Even so, the article makes a relevant contribution, since a rigorous analysis is necessary to obtain empirical evidence on the absence of a long-run relationship between oil revenue and HDI.

In the abstract, the authors blame the high volatility of oil prices for the poor performance of African countries in terms of sustainable development. However, some oil-producing countries have been successful in using fiscal policy rules and sovereign wealth funds to deal with volatility. See, for instance:

Taylor, A., Severson-Baker, C., Winfield, M., Woynillowicz, D., & Griffiths, M. (2004). When the government is the landlord. Economic rent, non-renewable permanent funds, and environmental impacts related to oil and gas developments in Canada. Report, Pembina Institute, Canada.

Therefore, countries’ failure to manage a large amount of oil revenues is not entirely due to price volatility. There must be something else.

In the conclusion, the authors argue that their results have important policy implications, and describe a complete set of required policies for taking advantage of a large amount of oil revenues. These policy measures to achieve sustainable development are well-known. The key question is then why these countries have not implemented these policies. The authors should address this question.

In my view, the bad performance of these countries in terms of HDI has to do with poor governance (low institutional quality). Indeed, poor governance results in inefficient management of oil revenues, which prevents these African countries from reaching higher HDI scores. As the authors themselves point out, oil revenues are not a curse per se, but the way in which countries manage them. In this respect, I miss some comments regarding governance indicators of these African countries (World Governance Indicators are available at https://datacatalog.worldbank.org/dataset/worldwide-governance-indicators).

The crucial role of institutional quality has been widely studied in the literature. The authors should survey this literature. See, for instance:

Mehlum, H., Moene, K., & Torvik, R. (2006). Institutions and the resource curse. The economic journal, 116(508), 1-20.

Ideally, some governance indicators should be introduced as explanatory variables in the empirical analysis.

As a minor concern, on pages 12-13, I guess the correlations in Table 3 were computed by the authors using Stata 15. Likewise, this software appears in a number of tables throughout the paper. Note that the software is not a source of figures in tables.

Author Response

Dear Reviewer,

Please see attached and below detailed point by point responses on the issues raised and the suggestions provided in the comment section and from the open review:

Reviewer 

SN

Summarized review point

Responses

Relevant Pages

1.        

Required more arguments to expand low HDI in crude oil-rich developing countries apart from price volatility

Further evidences were provided bothering on the World bank index referred to as, “Country policy and institutional assessment”. This was empirical attempt made to concretize why crude oil revenue has not produced the desired sustainable development level, as measured with the human development index (HDI).

See pages 11-12; 20-21; 27-29

2.        

Bothering on better well-furnished policy implications

The clarity required has been provided

See section 5, “Conclusion”, page 29

3.        

Introduction of governance some indicator and a suggestion to survey a supplied article title.

Again, “Country policy and institutional assessment (CPIA)” was introduced to explain why crude oil revenue has not yielded desired results in oil-rich developing countries.

See pages 11-12; 20-21; 27-29

4.        

Point bothering on data source referencing.

The sources of the data used have been correctly and rightfully cited.

See page 14 and other relevant sections where such corrections were made.

Round 2

Reviewer 1 Report

The article has improved a lot but in my opinion it is still missing:

The objectives of this study should be clearly stated to present the novelty of the study in the Introduction.

The originality and significance of the paper needs to be further clarified.

Author Response

Reviewer 1 Comments and Suggestions: Point by Point Response

Response on Reviewer 1: review report 2

Reviewer 1

SN

Summarized review point

Responses

Relevant Pages

1.        

The objectives of this study should be clearly stated to present the novelty of the study in the Introduction.

The objectives of the study have been clearly stated in the introductory part of the manuscript.

Pages 1-4: line 34-39, 46-69, and 112-140.

2.        

The originality and significance of the paper needs to be further clarified

The novelty of the study has been re-established in the manuscript to show that it has never been done, especially considering the period, places, methods and the objectives worked on.

Page 3-4: lines 112-140.

Reviewer 3 Report

This is a fine paper. I just have two minor comments. A third round is not necessary, since they can be easily fixed.

At the end of the abstract, I would write something like “This finding, therefore, requires an immediate fiscal intervention on spending on sustainable developing drivers such as education, health, agriculture cum adoption diversification policy and veritable supply-side policies could avert the possibility of negative effects. The absence of such policy interventions in these countries seems to be related to ineffective public institution or bad governance.” Note that good policies do not avoid ineffective public institution. On the contrary, better institutions lead to better policies.

On page 19, lines 666 and 667, country sub-groups are the same. May be it should say “…but bundled across sub-groups like “resource rich sub-Saharan Africa countries – oil exporters” and “resource rich sub-Saharan Africa countries – non-oil exporters”, amongst others.”

Results in sub-section 3.6 are really interesting. I encourage the authors to write a new paper on the connection between HDI and institutional indicators in resource-rich African countries. Note that the World Governance Indicators do contain statistical information for individual countries.

Author Response

Reviewer 3 Comments and Suggestions: Point by Point Response

Response on Reviewer 3: review report round 2

Reviewer 1

SN

Summarized review point

Responses

Relevant Pages

1.        

At the end of the abstract, I would write something like “This finding, therefore, requires an immediate fiscal intervention on spending on sustainable developing drivers such as education, health, agriculture cum adoption diversification policy and veritable supply-side policies could avert the possibility of negative effects. The absence of such policy interventions in these countries seems to be related to ineffective public institution or bad governance.” Note that good policies do not avoid ineffective public institution. On the contrary, better institutions lead to better policies.

The context provided by the reviewer is quite apt considering that it captures the results of the investigation and the minds of the authors. The abstract has been adjustment accordingly, with a few tweaks to the semantics. 

See page 1: Lines 25-31.

2.        

On page 19, lines 666 and 667, country sub-groups are the same. Maybe it should say “…but bundled across sub-groups like “resource rich sub-Saharan Africa countries – oil exporters” and “resource rich sub-Saharan Africa countries – non-oil exporters”, amongst others.”

The omission of the prefix “non” has been corrected to provide the right sense of the context.

See Section 3.6, paragraph 1, line 640, referencing the texts quoted by the reviewer.

3.        

Results in sub-section 3.6 are really interesting. I encourage the authors to write a new paper on the connection between HDI and institutional indicators in resource-rich African countries. Note that the World Governance Indicators do contain statistical information for individual countries

This suggestion has already been taken and might be worked on to substantiate our points of view on sustainable development viz-a-viz institutional factors, using HDI as a barometer of the former.

See 36: Lines 922-993
